# Genome-Wide Characterization and Expression Analysis of KH Family Genes Response to ABA and SA in *Arabidopsis thaliana*

**DOI:** 10.3390/ijms23010511

**Published:** 2022-01-03

**Authors:** Yanjie Zhang, Yu Ma, Ruiqi Liu, Guanglin Li

**Affiliations:** 1Key Laboratory of Ministry of Education for Medicinal Plant Resource and Natural Pharmaceutical Chemistry, Shaanxi Normal University, Xi’an 710119, China; camille@snnu.edu.cn (Y.Z.); yuma@snnu.edu.cn (Y.M.); 2College of Life Sciences, Shaanxi Normal University, Xi’an 710119, China; ruiqiliu@snnu.edu.cn

**Keywords:** KH family, expression, ABA and SA

## Abstract

K-homologous (KH) family is a type of nucleic acid-binding protein containing the KH domain and has been found to affect splicing and transcriptional regulation. However, KH family genes haven’t been investigated in plant species systematically. In this study, we identified 30 genes that belonged to the KH family based on HMM of the KH domain in *Arabidopsis thaliana*. Phylogenetic tree analysis showed that the KH family is grouped into three subgroups. Synteny analysis showed that *AtKH9* and *AtKH29* have the conserved synteny relationship between *A. thaliana* and the other five species. The *AtKH9* and *AtKH29* were located in the cytoplasm and nucleus. The seed germination rates of the mutants *atkh9* and *atkh29* were higher than wild-type after abscisic acid (ABA) and salicylic acid (SA) treatments. In addition, the expression of ABA-related genes, such as *ABRE-binding factor 2* (*ABF2*), *ABRE-binding factor 4* (*ABF4*), and *delta 1-pyrroline-5-carboxylate synthase* (*P5CS*), and an SA-related gene *pathogenesis-related proteins b* (*PR1b*) were downregulated after ABA and SA treatments, respectively. These results suggested that *atkh9* and *atkh29* mutants inhibit the effect of ABA and SA on seed germination. In conclusion, our results provide valuable information for further exploration of the function of KH family genes and propose directions and ideas for the identification and characterization of KH family genes in other plants.

## 1. Introduction

K-homologous (KH) family belongs to nucleic acid-binding protein containing KH domain, KH domain is first identified in human Heterogeneous Nuclear Ribonucleoprotein K (hnRNPK) protein [1,2]. The hnRNPK was discovered early in humans containing three conserved KH domains. The KH domain is composed of approximately 70 amino acids and is present in archaea, bacteria, and eukaryotes [1,3]. KH domain superfamily contains 14 families. KH domain can bind to RNA or DNA [4,5], and exists in different types of proteins, which play important roles in biological processes, including splicing, transcriptional regulation [6,7,8,9]. Now KH family genes haven’t been investigated in plant species systematically.

In plants, KH domain family genes have been shown to take part in developmental processes and adapt to environmental stresses. Several genes containing the KH domain were reported to involve in flowering time. For example, KH genes *FLOWERING LOCUS KH DOMAIN* (*FLK*) and *PEPPER* (*PEP*) interaction can regulate the expression of *FLOWERING LOCUS C* (*FLC*), which is the central inhibitor of flowering time in *Arabidopsis thaliana* [10,11,12,13]. KH domain protein *FLOWERING LOCUS Y* (*FLY*) controls flowering time by positively regulating the expression of *FLC* clade members [14]. In addition, the other KH gene *HUA ENHANCER 4* (*hen4*) mutant blooms earlier under abiotic stresses such as low temperature (16 °C) or long days [15,16]. Another KH domain gene, *REGULATOR OF CBF GENE EXPRESSION 3* (*RCF3*) [17], participates in the regulation of gene expression under heat stress. KH domain family genes also regulate the biogenesis of miRNA and pre-mRNA splicing. For example, *RCF3* is also a tissue-specific regulator of miRNA biogenesis by affecting the phosphorylation of miRNA biogenesis cofactor *HYPONASTIC LEAVES 1* (*HYL1*) in plants [17]. KH domain protein HOS5 with FIERY2/CTD phosphatase-like 1 and splicing factors and are important for pre-mRNA splicing in *A. thaliana* [18,19].

The phytohormone abscisic acid (ABA) plays an important role in many key processes of plant development and adaptation to biotic and abiotic stresses, such as regulating seed maturity, dormancy, germination and early seedling development [20,21,22]. ABA prevents seed germination and growth to protect plants from unfavorable conditions. Salicylic acid (SA) is a key molecule signal in plant growth and development. It can be produced when plants are exposed to drought, cold, or high osmotic pressure [23,24]. SA can promote seed germination under high salt concentration and osmotic pressure [25], while the highest concentration of SA inhibits the germination of maize seeds [26]. However, the KH family genes which are a response to ABA and SA are seldom investigated.

In this study, 30 KH family genes in *A. thaliana* were identified and their structures, locations, and expression profiles based on RNA-seq data were characterized. Also, the evolutionary relationship and synteny of KH family between *A. thaliana* and the other five species (*Brassica rapa*, *Cucumis sativus*, *Oryza sativa*, *Solanum lycopersicum*, and *Zea mays*) were analyzed. Among 30 KH family genes, *AtKH9* and *AtKH29*, the most conserved genes in subgroup I were selected to confirm subcellular localization and expression response to ABA and SA treatments. The results provide a foundation for further functional research on KH genes in *A. thaliana*.

## 2. Results

### 2.1. Identification and Characterization of KH Gene Family in Arabidopsis thaliana

Thirty KH family genes identified in *A. thaliana* were named from *AtKH1* to *AtKH30*. All 30 KH family genes were scattered on five chromosomes of *A. thaliana* (Appendix A). Among them, there are the most genes on chromosome 5 (nine genes) and the least on chromosomes 1 (four genes) and 4 (four genes). Their CDS length ranged from 465 to 2766 bp, and the length of the corresponding encoded amino acid ranged from 153 to 922. The pI of these proteins ranged from 4.16 to 9.71 and their MW ranged from 17.376 to 99.566 kDa (Appendix A).

A phylogenetic tree was constructed to analyze the possible evolutionary relationship of KH family in *A. thaliana*. The 30 genes were divided into three subgroups (subgroup I, II, and III) which contains six, nine, and fifteen members, respectively (Figure 1). And, the exon-intron structures of the members in the same subgroup are similar and most of them contained 5–8 introns (Figure 1). In subgroup I, *AtKH1*, *AtKH12*, and *AtKH21* were composed of six exons, while *AtKH9* and *AtKH29* were composed of seven exons (Figure 1). In addition, the number of KH domains in different subgroups was different (Figure 1, Appendix A). Among them, the number of KH domains in subgroup I, II, and III (1, 2–3, and 3–5) indicated that genes in the same subgroup have more similar gene structures. In all KH family proteins, a total of twelve conserved motifs were found. Among them, subgroups I mainly included four motifs (Motif 1, 3, 5, and 8), subgroups II is mainly contained five motifs (Motif 1, 2, 4, 9, and11), and III mainly included three motifs (Motif 1, 2, and 4). The *cis*-elements in the promoter region of KH family in *A. thaliana* were analyzed, and the result revealed that these *cis*-elements mainly include three types: biotic/abiotic stress responses (light-responsiveness and drought-inducibility), plant growth and development (meristem expression), and phytohormone responses (ABA-responsiveness). Light-responsive *cis*-elements exist in the promoter region of all KH family genes (Appendix A).

### 2.2. Phylogenetic Analysis of KH Family

For a more detailed study of the KH domain, we analyzed the protein sequences of six species based on raw HMM of the KH domain downloaded from Pfam. As a result, 30 KH family genes in *A. thaliana*, 42 KH family genes in *Brassica rapa*, 28 KH family genes in *Cucumis sativus*, 34 KH family genes in *Oryza sativa*, 45 KH family genes in *Solanum lycopersicum*, 62 KH family genes in *Zea mays* were obtained, respectively (Appendix A). All of these 241 genes were used to construct the phylogenetic tree of the KH family. As a result, all 241 KH family genes were divided into three subgroups (Figure 2a). Each subgroup contained all six species, revealing that all these KH family genes come from the same ancestor. A total of 233 of these 241 genes were found to contain similar motif sequences with a high proportion of G in 19th and 22nd positions in the KH motif sequence (GEVTVRJLVPSSKVGSII**G**KG**G**STIKRJREETGARIRI), which may be the conserved sequence motif in KH domain [27] (Figure 2b).

### 2.3. Synteny Analysis of KH Family

Synteny analysis between *A. thaliana* and the other five species was used to study the evolutionary relationship of KH family genes (Figure 3 and Appendix A). We found that twenty-two gene pairs were found in synteny blocks between *A. thaliana* and *B. rapa* (*AtKH3*-*BrKH27*, *AtKH4*-*BrKH16*, *AtKH6*-*BrKH33/41*, *AtKH8*-*BrKH10/37*, *AtKH9*-*BrKH1/7*, *AtKH10*-*BrKH2/35*, *AtKH18*-*BrKH27*, *AtKH19-BrKH14/24*, *AtKH20*-*BrKH15/18*, *AtKH24*-*BrKH22*, *AtKH28*-*BrKH6/21*, and *AtKH29*-*BrKH5/17/25/30*); ten pairs of genes were found in synteny blocks between *A. thaliana* and *C. sativus (AtKH3*-*CsKH12*, *AtKH4*-*CsKH5*, *AtKH8*-*CsKH9*, *AtKH9*-*CsKH20/23*, *AtKH14*-*CsKH25*, *AtKH20*-*CsKH10*, *AtKH18*-*CsKH12*, *AtKH19*-*CsKH1*, and *AtKH29*-*CsKH27*). Between *A. thaliana* and *O. sativa*, *S. lycopersicum*, *Z. mays*, there were two (*AtKH9*-*OsKH6* and *AtKH29*-*OsKH10*), six (*AtKH6*-*SlKH3/11*, *AtKH8*-*SlKH15*, *AtKH9*-*SlKH38*, *AtKH24*-*SlKH3*, and *AtKH29*-*SlKH44*), and one (*AtKH29*-*ZmKH32*) gene pairs were found in synteny blocks, respectively. Some genes were found in synteny blocks between *A. thaliana* and other two species, such as *AtKH4*, *AtKH18*, *AtKH19*, and *AtKH20* in *B. rapa* and *C. sativus*; *AtKH6* and *AtKH24* in *B. rapa* and *S. lycopersicum*. More than this, *AtKH9* has gene pairs with the KH family in synteny blocks between *A. thaliana* and the other four species except for *Z. mays*, and *AtKH29* has gene pairs with the KH family in synteny blocks between *A. thaliana* and the other five species. These results indicated that compared with other KH family genes, *AtKH29* and *AtKH9* may be more conserved.

### 2.4. Expression of KH Family Genes in Different Stages during Growth and Development

The RNA-seq data of the wild-type Col-0 was downloaded to detect the time-specific expression of the KH family genes in the growth and development of *A. thaliana* (Figure 4). *AtKH6*, *AtKH8*, *AtKH16*, and *AtKH21* had similar patterns during growth and development, all of them have the highest expression in the flowing stage, suggesting that *AtKH6/8/16/21* may be related to flower development regulation. Interestingly, we found that with the growth and development of plants, the expression level of *AtKH12* will continue to increase and *AtKH9* will continue to decrease though both of them were included in the same subgroup. In contrast, *AtKH3* and *AtKH9* showed the highest expression levels in the seed stage, decreased expression levels in the leaf stages (relatively consistent expression levels in the leaf stage 1 and leaf stage 2), and further decreased in the flower stage and senescence stage though they belonged to two different subgroups. In addition, the other 15 genes had the highest expression in the seed stage. All the results suggest that KH family genes may perform different functions in growth and development.

### 2.5. Expression of KH Family Genes under ABA and SA Treatments

Expression data of *A. thaliana* was downloaded to analyze changes of expression under ABA and SA treatments. As shown in Figure 5a, the expression levels of *AtKH2/5/9/13/17/21/25/28/29/30* after ABA treatment were increased at 1 h, among which the expression of *AtKH2/17/25/29/30* kept stable at 3 h and the expression of *AtKH5/9/13/21/28* decreased at 3 h. In contrast, the expression of *AtKH4/12/15/19/20/26* decreased at 1 h and then increased at 3 h. Also, the expression of *AtKH8/14/27* was insignificantly changed at 1 h and decreased at 3 h. Expression of other ten genes (*AtKH1/3/6/7/10/11/16/18/22/23*) was decreased after ABA treatment. Expression of KH family genes after SA treatment was complicated (Figure 5b). For example, the expression levels of *AtKH1/4/10/11/14/21/22/23/28* were decreased after SA treatment at 6 h, 12 h, and 24 h, while expression of *AtKH14/21/28* recovered the expression at 48 h. The expression of some genes (*AtKH2/3/16/17/27*) was generally increased after SA treatment. Other genes have the highest expression at a specific time point after SA treatment. These results suggested that KH family genes have different responses after ABA and SA treatments.

### 2.6. Function Analysis of AtKH9 and AtKH29

To further verify the function the of KH family, we selected the most conserved two genes in subgroup I, *AtKH9/29* for the next analysis. The leaves transformed with the control vector (35S:GFP) showed the fluorescence of GFP was in the plasma membrane and nucleus (Figure 6), and the leaves transformed with *AtKH9/29*-GFP showed the fluorescence signals of *AtKH9/29* appeared in the plasma membrane and nucleus.

To verify the changes in the expression of *AtKH9/29* after the analysis of ABA and SA treatments, 15-day-old *A. thaliana* seedlings were collected as samples after ABA and SA treatments. The expression of *AtKH9/29* genes increased at 3 h after ABA treatment, indicating that *AtKH9/29* might respond to ABA regulation. For SA treatment, the expression of *AtKH9/29* genes was increased at 12 h and peaked at 24 h (Figure 7). These results indicated that *AtKH9/29* can respond to ABA and SA.

In order to further determine the functions of *AtKH9* and *AtKH29*, *atkh9* and *atkh29* mutants were sown containing ABA and SA with different concentrations. We found that the 4-day-old seed germination rates of *atkh9* and *atkh29* were higher than wild-type under ABA and SA treatments (Figure 8). The germination rates of *atkh9/29* and wide-type under ABA and SA treatments changed most obviously on the 6 d, and they remained stable on the 8 d (Appendix A). In conclusion, *atkh9/29* is less stressed than wild-type by ABA and SA.

Then the expression of ABA and SA-related genes were detected in the mutants under ABA and SA treatments, respectively. We found that the expression of ABA-related genes *ABRE-binding factor 2* (*ABF2*), *ABRE-binding factor 4* (*ABF4*) [28], and *delta 1-pyrroline-5-carboxylate synthase* (*P5CS*) [29] in the mutants were lower than in wild-type after ABA treatment (Figure 9a). *ABF2*, *ABF4*, and *P5CS* are marker genes of the ABA dependent pathway. Also, the expression level of SA response gene *pathogenesis-related proteins b* (*PR1b*) [30,31] in the mutants under SA stress was lower than in wild-type (Figure 9b). *PR1b* is the marker gene of SA dependent pathway. These results showed that *atkh9/29* was less insensitive than wild-type after ABA and SA treatments.

## 3. Discussion

The K-homologous (KH) family is a type of nucleic acid-binding protein containing the KH domain, which plays important role in the development and response to environmental stress [1,18,32]. In *A. thaliana* now there are only five of thirty KH genes including *FLK*, *PEP*, *FLY*, *HEN4* and *RCF3* [10,11,12,13,14,15,16,17], whose functions have been reported. However, the classification, expression, and function of KH family genes haven’t been investigated in *A. thaliana* systematically. In this study, we firstly identified and classified KH family genes, and analyzed their genomic characteristics including the location on the chromosome, CDS and protein sequence length, pI and MW, conserved motifs, cis-elements in the promoter region, phylogenetic tree, synteny. Then we revealed the expression pattern of these KH genes in both the growing and development of *A. thaliana* and response to ABA and SA. Finally, the two conserved KH genes, *AtKH9* and *AtKH29*, were used to investigate the function of KH gene including their subcellular localization, seed germination rates under ABA and SA treatments by experiment method.

In this study, we identified 30 genes that belonged to the KH family based on HMM of the KH domain in *A. thaliana*. Compared to the 26 KH family genes identified by Lorković’s group [33], we have found five of 30 KH family genes in *A. thaliana* which have not been identified previously. Phylogenetic tree analysis showed that these KH family genes can be divided into three subgroups by using 30 KH family genes from *A. thaliana*. The gene structures, the number of KH domains, and the type of motifs in each subgroup are relatively similar. Some genes of the KH family that have been reported to be functional in plants have also been discovered in our study. The three genes that regulate the flowering time of plants by interacting with *FLK*, *PEP*, and *FLY* [10,11,12,13,14] all belong to subgroup II, while the two genes involved in plant response to abiotic stress (*HEN4* and *RCF3*) [15,16,17] belong to subgroup III, this provides new ideas for the next step of studying the function of KH family genes. Most KH family genes have more than one KH domain, which is consistent with previous studies [33].

There are many tandem duplications and fragment duplications in the *A. thaliana* genome, which are important factors affecting the evolution of gene families [34], another study showed that segment duplication appears to be the main form of evolution of the expansion gene superfamily in *A. thaliana* and *Glycine max* [35]. The analysis of synteny revealed that there is no tandem duplication pair and only one segmental duplication pair in *A. thaliana*, which suggests that there is no expansion of KH family in *A. thaliana* by tandem duplications or segmental duplications, this may be related to the structure of the KH domain [3]. Then, a total of 241 KH family genes were obtained in six species, almost all of these 241 KH family genes contain a similar motif (Figure 1). Synteny analysis between *A. thaliana* and the other five species revealed that *AtKH9* and *AtKH29* may be the most conserved genes in KH family.

The *cis*-elements in the promoter region of *A. thaliana* were also predicted, all of which can be divided into three types: biotic/abiotic stress responses, plant growth and development, and phytohormone responses. All of KH family genes contain light-responsive *cis*-elements (GT1-motif, GGTTAA), indicating that these KH family genes might contribute to the response of *A. thaliana* to light. Also, 24 of 30 KH family genes contain ABA-responsive *cis*-elements (ABRE, AACCCGG), which indicated some KH family genes such as *AtKH1*, *AtKH9* can respond to ABA through these *cis*-elements. ABA and SA are the important hormones in plants, which are involved in diverse processes, such as germination and adaptation to different stresses [20,21,22,23,24,25,26].

In *A. thaliana*, only one KH family gene *RCF3* was reported in response to heat stress [17]. In order to search for the KH family genes response to ABA and SA treatments. RNA-seq data are used to analyze the expression pattern KH family genes in *A. thaliana*. The result showed that *AtKH5/9/12/13/15/17/20/25/26/29/30* are differentially expressed, which indicated these KH genes may respond to ABA and SA, respectively. In order to validate these results further, we chose *AtKH9* and *AtKH29* to investigate the expression pattern of *AtKH9* and *AtKH29* under ABA and SA treatments using qRT-PCR. The result also showed that the expression of *AtKH9/29* genes was increased in both treatments (Figure 7). The result of RNA analysis and qRT-PCR are all indicated that *AtKH9/29* may respond to ABA and SA.

In order to investigate the function of *AtKH9* and *AtKH29* further, the location of two genes in *A. thaliana* was firstly determined, and we found that that they are located in the cytoplasm and nuclear where they may function through interaction with RNA, DNA, or protein. Then we tested the ratio of seed germination under different concentration of ABA and SA, the result showed that the higher the concentrations of ABA and SA, the more obvious the inhibitory effect on seed germination, while the seed germination ratios of *AtKH9/29* mutants (*atkh9/29*) were higher than wild-type. So, the *AtKH9/29* is sensitive to ABA and SA treatments. WRKY transcription factors (TFs) are key regulators of many plant processes, including the responses to biotic and abiotic stresses, and seed germination, which may play roles in regulating plant responses to the ABA, and involved in ABA signaling include well-known ABA-responsive genes (*ABF2*, *ABF4*) [28]. Accumulation of *P5CS* transcripts is regulated in a tissue-specific manner and inducible by ABA [29]. *PR1b* is marker gene of the SA-dependent defense pathway, was significantly up-regulated in response to SA [30]. The marker genes involved in the ABA signaling pathway including *ABF2*, *ABF4*, and *P5CS*, and SA signaling pathway including *PR1b* were also detected, and it was found that the expression level of these genes in wild-type was higher than that of *atkh9/29* under the same treatment, respectively. These results suggested that *AtKH9* and *AtKH29* play important roles in response to ABA and SA. In conclusion, our study may provide better help to understand the source and function of KH family genes.

## 4. Materials and Methods

### 4.1. Identification and Characterization of KH Gene Family in Arabidopsis thaliana

The genome sequences and annotation files of *Arabidopsis thaliana*, *Brassica rapa*, *Cucumis sativus*, *Oryza sativa*, *Solanum lycopersicum*, and *Zea mays* were downloaded in Ensembl Plants (release-51, http://plants.ensembl.org/, accessed on 10 January 2021). CDS sequences and protein sequences were extracted from the genome sequence and annotation file of each species using TBtools (v1.0971) [36].

To find KH family sequences in these six species, the Hidden Markov Models (HMM) of KH domain (PF00013) was downloaded from Pfam (http://pfam.xfam.org/, accessed on 10 January 2021) [37], and then the protein sequences of six species were applied as query sequences to search putative KH family sequences against PF00013 domain using HMMER (v3.3.2) [38] with *E*-value < 0.01. Subsequently, protein sequences of the putative KH family were uploaded to Pfam to confirm the existence of KH domain. Finally, the gene sequence corresponding to each protein sequence of KH family was extracted, and each gene of KH family was named according to the position of the gene in the chromosome.

The isoelectric point (pI) and molecular weight (MW) of each KH sequence were calculated in Expasy (http://web.expasy.org/, accessed on 10 January 2021). MEME (v5.4.1, https://meme-suite.org/meme/, accessed on 10 January 2021) [39] was used to investigate conserved motifs in KH family and the location of KH domain was obtained from Pfam. The exon-intron structures for KH family were graphed by TBtools. The PlantCARE (http://bioinformatics.psb.ugent.be/webtools/plantcare/html/, accessed on 10 January 2021) was used to predict cis-elements in the promoter region.

### 4.2. Multiple Sequence Alignment and Phylogeny of KH Family

Full-length protein sequences of all six species were aligned using the online tool mafft (https://www.ebi.ac.uk/Tools/msa/, accessed on 10 January 2021) and then exported the results to MEGA X [40] to construct maximum likelihood (ML) phylogenetic trees with bootstrap analysis (1000 replicates). Online tool iTOL (https://itol.embl.de/, accessed on 10 January 2021) was used to visualize the phylogenetic tree.

### 4.3. Synteny Analysis and Expression Analysis Based on RNA-Seq Data

TBtools was used to analyze the synteny within *A. thaliana* and between *A. thaliana* and other five species with *E*-value < 1 × 10^−10^. The expression (normalized by the GCOS method, TGT value of 100) during different stages of growth and development were downloaded from ePlant (http://bar.utoronto.ca/eplant/, accessed on 10 January 2021) [41]. For convenience, the growth and development process of *A. thaliana* were divided into five stages: seed stage, leaf stage 1 (1–7 leaves), leaf stage 2 (8–14 leaves), flowering stage, and senescence stage [42]. And, the expression counted by FPKM (Fragments Per Kilobase of exon model per Million mapped fragments) values, under ABA and SA treatments were downloaded from the Arabidopsis RNA-seq Database (http://ipf.sustech.edu.cn/pub/athrdb/, accessed on 10 January 2021, Accession number: PRJNA513154 and PRJNA509414) [43]. The Z-Score was calculated for heatmap plotting.

### 4.4. Plant Materials and Growing Conditions

The mutants of *atkh9* (SALK_056508) and *atkh29* (SALK_000684) were from Arashare (https://www.arashare.cn/index/, accessed on 10 January 2021), and verified (Appendix A). Wild-type Col-0, *atkh9*, and *atkh29* seeds were surface-disinfected in 0.01 % sodium hypochlorite, and the seeds were planted on a semi-strength MS agar medium. The seeds were vernalized at 4 °C for 48 h. The plants were grown in an incubator at 21 °C with a light intensity of 5000 Lx with 16 h-light/8 h-darkness.

### 4.5. Subcellular Localization

Our bioinformatic analysis showed that *atkh9* and *atkh29* are more conservative compared to other KH family genes. To examine the localization of *AtKH9/29* in plants, Agrobacterium tumefaciens GV3101 containing 35S:GFP and 35S:*AtKH9/29*-GFP construction was resuspended in infiltration buffer (10 mM MgCl_2_, 10 mM MES, pH 5.7-6.0, 200 μM AS) before infiltration into 30-days-old leaves of *Nicotiana benthamiana*. Two days later, the fluorescence of GFP was observed using a confocal microscope (Leica TCS SP8) with 488 nm of excitation wavelength and 495-530 nm of emission wavelength. Chloroplasts were observed with 630 nm of excitation wavelength and 643–730 nm of emission wavelength [44].

### 4.6. Materials of AtKH9/29 Genes Expression Analysis and Seed Germination

Wild-type Col-0 seeds were surface-disinfected in bleach, and the seeds were surface-disinfected in 0.01 % sodium hypochlorite, and the seeds were planted on semi-strength MS agar medium The seeds were vernalized at 4 °C for 48 h. The plants were grown in an incubator at 21 °C with a light intensity of 5000 Lx with 16 h-light/8 h-darkness. The 15-day seedlings to collect samples at 0 h, 1 h, 3 h after ABA treatment, and 0 h, 6 h, 12 h, 24 h, and 48 h after SA treatment to detect *AtKH9/29* expression.

Wild-type Col-0 and mutants (*atkh9*, *atkh29*) *A. thaliana* seeds were surface-disinfected in bleach, and the seeds were surface-disinfected in 0.01 % sodium hypochlorite, and the seeds were planted on semi-strength MS agar medium supplemented with containing 0.25 μM, 0.5 μM, 1 μM ABA and 50 μM, 100 μM, 200 μM SA. The seeds were vernalized at 4 °C for 48 h. The plants were grown in an incubator at 21 °C with a light intensity of 5000 Lx with 16 h-light/8 h-darkness. Each genotype was sown with 54 seeds of wild-type Col-0 and mutants (*atkh9*, *atkh29*) with three repetitions. The seed germination rates were counted within 3 d, 4 d, 6 d, 8 d, 10 d after the vernalization [45].

### 4.7. ABA and SA-Related Gene Expression Analysis

Wild-type Col-0, mutants *atkh9*, and *atkh29* seeds were surface-disinfected in bleach, and the seeds were sown in semi-strength MS agar medium supplemented with 0.5 μM ABA and 100 μM SA. After 15 d, Col-0, *atkh9* (SALK_056508), and *atkh29* (SALK_064329) seedlings were used to extract total RNA. Total RNA was extracted using Plant RNA Kit (Beibei, Zhengzhou, China) following the instructions. Two μg RNA was used for cDNA synthesis using HiScript^®^ III RT SuperMix for qPCR (+gDNAwiper) (YEASEN, Shanghai, China). The cDNA was used to determine the expression of ABA and SA-related genes *ABF2*, *ABF4*, *P5CS*, *PR1b* [28,29,30,31]. qRT-PCR was performed on the CFX96 Touch real-time PCR detection system using SYBR Green (YEASEN, Nanjing, China) at 95 °C for 3 min, 95 °C for 15 s, and 60 °C for 20 s in 45 cycles (Bio-rad, Berkeley, CA, USA). Atactin2 was used as an internal control. The relative expression level of genes was calculated using the 2^−ΔΔ^Ct method with three biological replicates. The primers are listed in Appendix A.

## Figures and Tables

**Figure 1 ijms-23-00511-f001:**
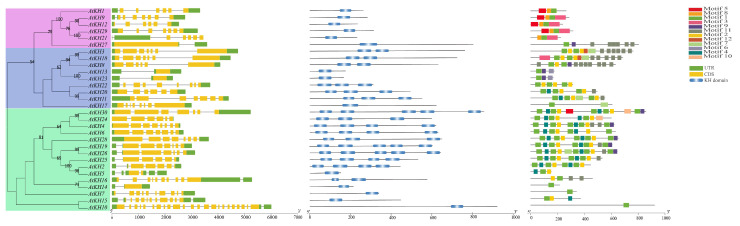
The phylogenetic tree, gene structure, KH domain location, and position of conservative motifs of 30 *Arabidopsis thaliana* KH family genes. From left to right were the unrooted phylogenetic tree, gene structure, position of KH domain and position of conservative motif. The axis below indicated the length of the CDS and protein.

**Figure 2 ijms-23-00511-f002:**
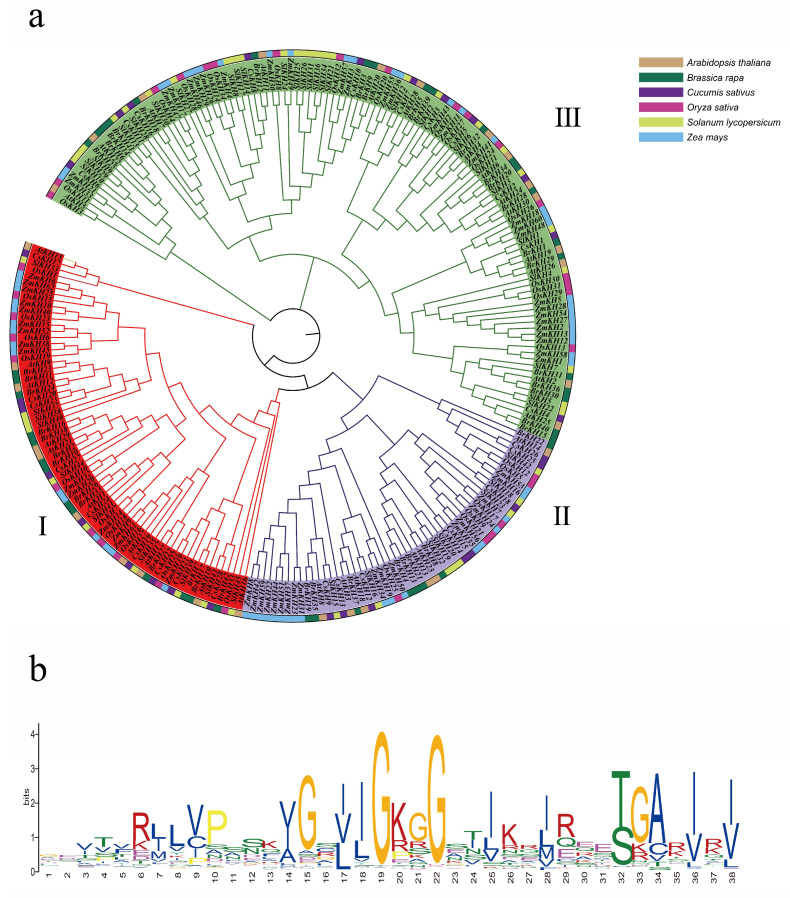
The phylogenetic tree and the main motif of KH family genes in six species. (**a**) The phylogenetic tree was constructed by hypothetical KH domain genes in six species. The outer layer showed the KH domain genes from six different species in six different colors, and three colors of the inner layer were used to mark the three subgroups. (**b**) The main motif of the identified KH genes.

**Figure 3 ijms-23-00511-f003:**
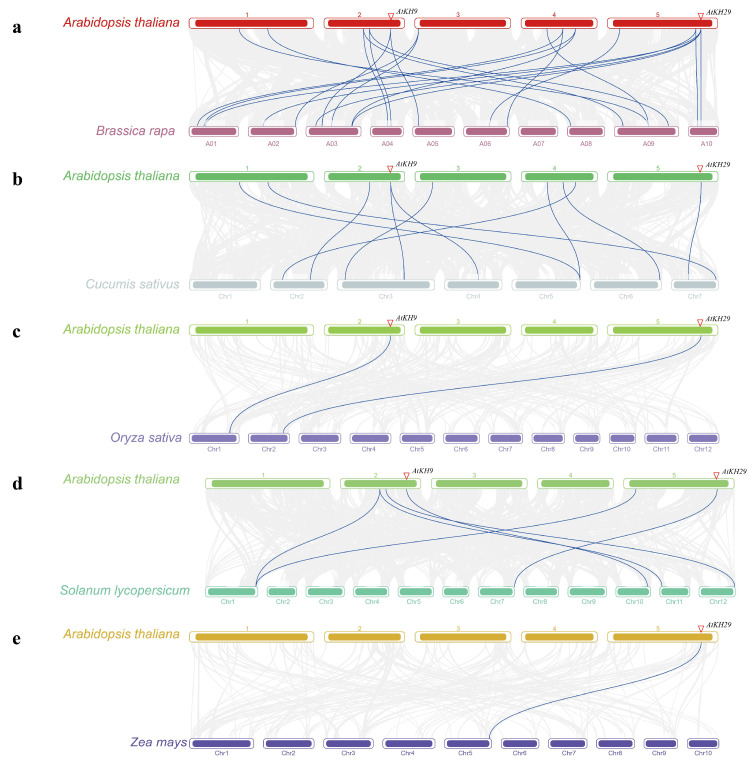
Synteny analysis of KH domain genes in *Arabidopsis thaliana* and other five species. The gray line represented all genes with synteny between the two species, and the synteny of KH domain genes was represented by the blue line. (**a**) Synteny between *A. thaliana* and *Brassica rapa*. (**b**) Synteny between *A. thaliana* and *Cucumis sativus*. (**c**) Synteny between *A. thaliana* and *Oryza sativa*. (**d**) Synteny between *A. thaliana* and *Solanum lycopersicum*. (**e**) Synteny between *A. thaliana* and *Zea mays*.

**Figure 4 ijms-23-00511-f004:**
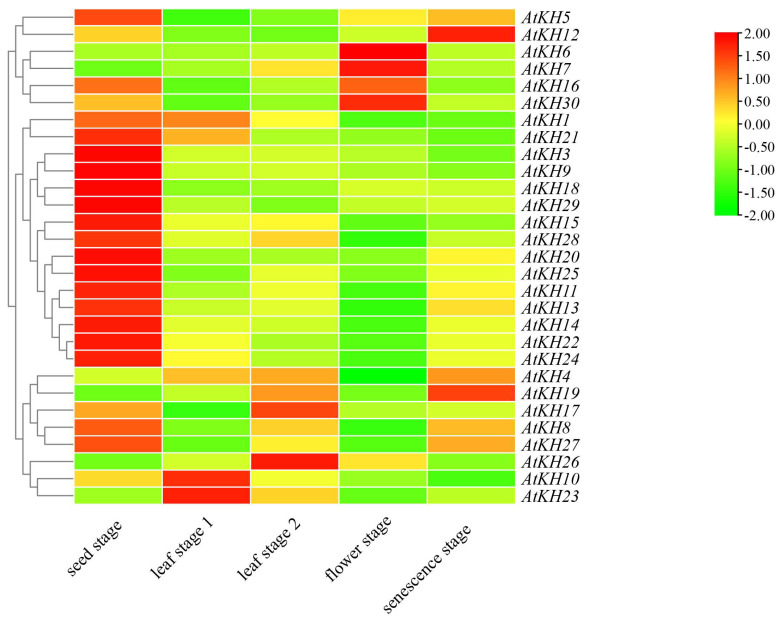
The expression of *Arabidopsis thaliana* KH domain genes during the growth and development. The growth and development stage of *A. thaliana* was artificially divided into five stages, namely seed stage, leaf stage 1 (leaf number 1–7), leaf stage 2 (leaf number 8–14), flower stage, and senescence stage.

**Figure 5 ijms-23-00511-f005:**
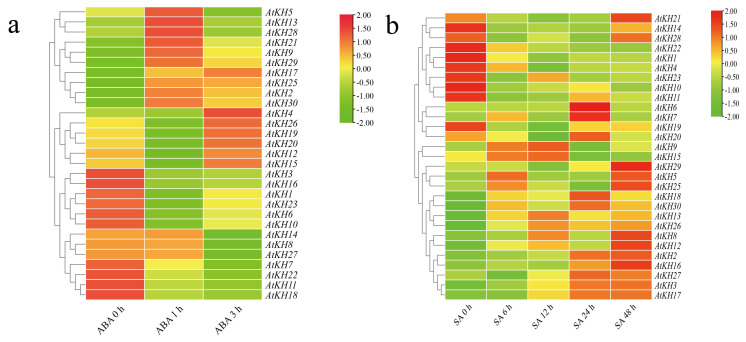
The expression of *Arabidopsis thaliana* KH domain genes under treatments of ABA and SA. (**a**) The expression of 30 *A. thaliana* KH genes at 0 h, 1 h, and 3 h after treatment with ABA in the wild-type Col-0. (**b**) The expression of 30 *A. thaliana* KH genes at 6 h, 12 h, 24 h, and 48 h after SA treatment.

**Figure 6 ijms-23-00511-f006:**
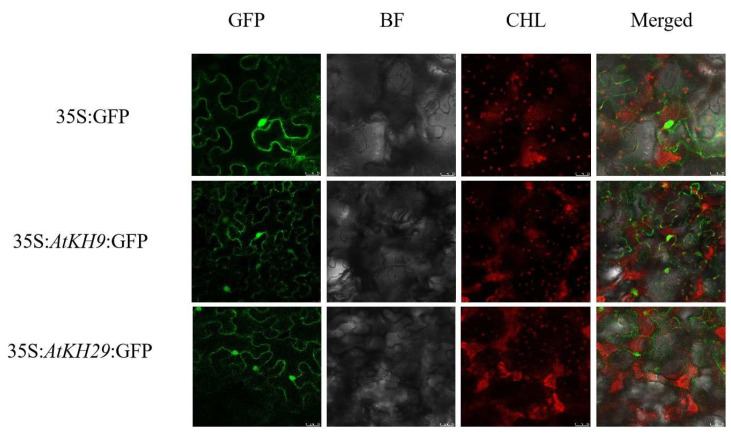
Subcellular location of *AtKH9* and *AtKH29* in *Nicotiana benthamiana*. GFP: Green fluorescent protein; BF: Bright-field images; CHL: Chloroplast fluorescence; Merge: Merged images.

**Figure 7 ijms-23-00511-f007:**
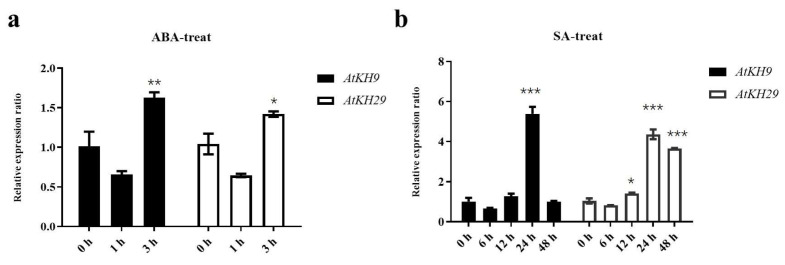
The expression levels of *AtKH9/29.* (**a**) *AtKH9/29* expression levels at 0 h, 1 h, 3 h after ABA treatment. (**b**) *AtKH9/29* expression levels 0 h, 6 h, 12 h, 24 h and 48 h after SA treatment. Error bars indicated SD. *p* value (* *p* < 0.05, ** *p* < 0.01, *** *p* < 0.001) was calculated using one-way analysis of variance and Tukey’s test for statistics.

**Figure 8 ijms-23-00511-f008:**
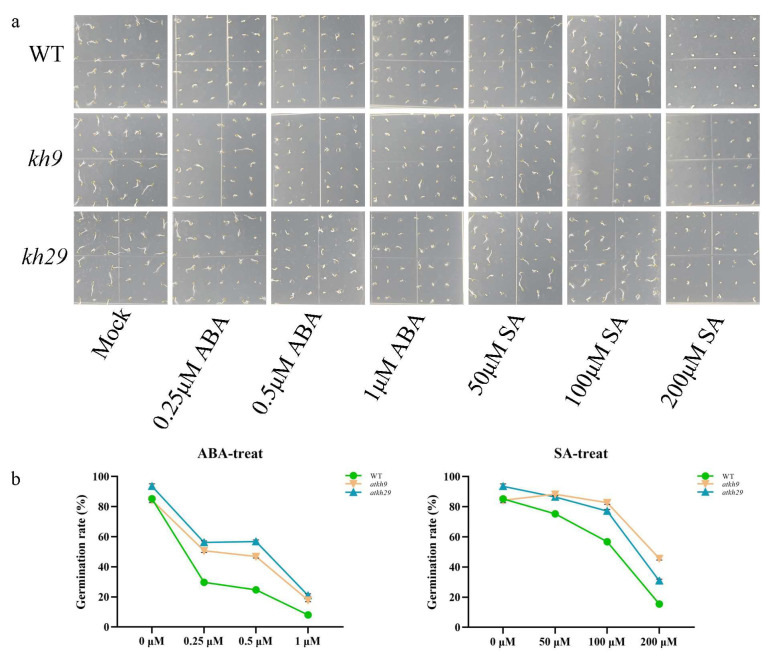
The seed germination rates of *atkh9* and *atkh29* under ABA and SA treatments. (**a**) Picture of 4 d seedlings of wild-type, *atkh9*, and *atkh29* on the medium containing 0.25 μΜ, 0.5 μΜ, 1 μΜ ABA, and 50 μΜ, 100 μΜ, 200 μΜ SA. (**b**) Statistics of the germination rates of wild-type, *atkh9* and *atkh29*. Error bars indicated SD.

**Figure 9 ijms-23-00511-f009:**
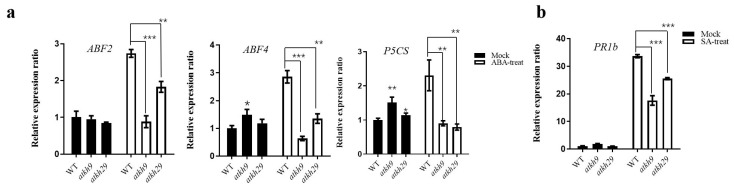
The expression of ABA and SA related genes of the wild type, *atkh9* and *atkh29* mutants under ABA and SA treatments for 10 d. (**a**) The expression of ABA-related genes *ABF2*, *ABF4*, and *P5CS*. (**b**) The expression of SA-related gene *PR1b*. The histogram indicated the average value of the experiment with three replicates. Error bars indicated SD. *p* value (* *p* < 0.05, ** *p* < 0.01, *** *p* < 0.001) was calculated using one-way analysis of variance and Tukey’s test for statistics.

## Data Availability

Publicly available datasets were analyzed in this study. These data can be found here: [http://ipf.sustech.edu.cn/pub/athrdb/]; [http://bar.utoronto.ca/eplant/], accessed on 10 January 2021.

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
