# Peer review of "Genome-Wide Characterization and Expression Analysis of KH Family Genes Response to ABA and SA in Arabidopsis thaliana"

_ijms, 2022, doi:10.3390/ijms23010511_

Round 1

Reviewer 1 Report

This manuscript describes the characterization and functional analysis of the Arabidopsis KH family, with interesting results showing that AtKH9 and AtKH29 may be involved in seed germination via abscisic acid and salicylic acid signaling. However, the reviewer cannot recommend accepting this paper because of the following serious issues with the description and the analyses.

Major points

1) The figures in the paper are not inserted in the Manuscript file, is this following with the journal format? Also, in Figure 1, the font for the gene names is too small and difficult to distinguish. Please consider making the font larger or using a different notation.

2) Does synteny mean that multiple genes co-localize? This paper shows only the KH genes when discussing synteny. Since there is no information on the surrounding genes, I cannot judge whether the synteny discussed in the paper is appropriate or not. Also, why are the combinations of genes that are supposed to have synteny often not the closest relationships in the phylogenetic tree in Figure 1? For example, AtKH9-BrKH1/7 is described to have synteny (Line 69), but in Figure 1, AtKH9 is close to BrKH23. Since genes that have synteny have the same origin, shouldn't they be classified as close lineages in gene sequences?

3) (Lines 94-97) "The results showed that among the 30 KH domain genes in A. thaliana, only AtKH3 and AtKH18 have repetitive gene pairings ( Figure S1), which indicates that the KH family in A. thaliana may not have extensive gene duplications like other gene families." 
If gene duplication occurs only for the KH gene, synteny conservation does not occur. Therefore, isn't this discussion inappropriate?

4) The gene expression analysis is only RNA-seq analysis from public databases. At least for AtKH9/29, which is the focus of the paper, quantitative RT-PCR analysis should be performed.

5) In the analysis of seed germination rate, the dose-dependency of ABA and SA and the time-course analysis should be investigated. Without time-course analysis, it is not possible to tell whether seed germination is delayed or does not occur.

6) Are the genes discussed in Figure 8 genes that are involved in seed germination? Is PR1b a pathogenesis-related gene and has not been reported to be involved in seed germination?

Minor Points
7) In line 19, "inhibit the effect of ABA and SA on seed germination." should be removed.

Author Response

Comments and Suggestions for Authors

This manuscript describes the characterization and functional analysis of the Arabidopsis KH family, with interesting results showing that AtKH9 and AtKH29 may be involved in seed germination via abscisic acid and salicylic acid signaling. However, the reviewer cannot recommend accepting this paper because of the following serious issues with the description and the analyses.

Major points

1) The figures in the paper are not inserted in the Manuscript file, is this following with the journal format? Also, in Figure 1, the font for the gene names is too small and difficult to distinguish. Please consider making the font larger or using a different notation.

Response 1: Thank you so much for your suggestion on our manuscript, IJMS now accepts free format submission, in order to avoid errors in the text of the manuscript after inserting the picture, we  store all pictures in a pdf file temporarily. We adjusted the gene names in Figure 1 as much as possible for easy viewing. Due to the revision of the manuscript content, we displayed the final information in Figure 2.

2) Does synteny mean that multiple genes co-localize? This paper shows only the KH genes when discussing synteny. Since there is no information on the surrounding genes, I cannot judge whether the synteny discussed in the paper is appropriate or not. Also, why are the combinations of genes that are supposed to have synteny often not the closest relationships in the phylogenetic tree in Figure 1? For example, AtKH9-BrKH1/7 is described to have synteny (Line 69), but in Figure 1, AtKH9 is close to BrKH23. Since genes that have synteny have the same origin, shouldn't they be classified as close lineages in gene sequences?

Response 2: Thank you so much for your suggestion on our manuscript, synteny means a set of loci in two different species which is located on the same chromosome in each (not necessary in the same order), is used in genetics to indicate the presence of two or more loci on the same chromosome. There are some descriptive errors in our previous manuscripts. It is not that there is a synteny relationship between individual genes, but that there is a synteny relationship between a set of genes and another set of genes, which is called a synteny block, and we just our KH family genes in these synteny blocks. We have now corrected these errors and show an example of AtKH9-BrKH1 to display the synteny blocks in Figure S3 , the blue line refers to AtKH9-BrKH1, and the other colored lines refer to the surrounding genes of AtKH9 and BrKH1. As can be seen from Figure S3, in Arabidopsis thaliana and Brassica rapa, these genes do constitute a synteny block. Both synteny analysis and phylogenetic tree analysis can describe the evolutionary relationship of genes. Synteny analysis focuses more on the relationship between a set of genes and another set of genes, while phylogenetic tree analysis focuses more on the relationship between one gene and another gene. Generally, the genes in synteny block will be close in the phylogenetic tree. The existence of different numbers of domains may be one of the reasons why they are far away on the phylogenetic tree.

3) (Lines 94-97) "The results showed that among the 30 KH domain genes in A. thaliana, only AtKH3 and AtKH18 have repetitive gene pairings ( Figure S1), which indicates that the KH family in A. thaliana may not have extensive gene duplications like other gene families."

If gene duplication occurs only for the KH gene, synteny conservation does not occur. Therefore, isn't this discussion inappropriate?

Response 3: Thank you so much for your suggestion on our manuscript, this statement is indeed inappropriate. For the sake of rigor, we have deleted this sentence in the new manuscript.

  • The gene expression analysis is only RNA-seq analysis from public databases. At least for AtKH9/29, which is the focus of the paper, quantitative RT-PCR analysis should be performed.

Response 4: Thank you so much for your suggestion on our manuscript, quantitative RT-PCR analysis of AtKH9/29 under ABA and SA was shown in Figure 7 in the new manuscript. To verify the changes in the expression of AtKH9/29 after the analysis of ABA and SA treatment, 15-day-old A. thaliana seedlings were collected as samples after ABA and SA treatment. The expression of AtKH9/29 genes increased at 3 h after ABA treatment, indicating that AtKH9/29 might respond to ABA regulation. For SA treatment, the expression of AtKH9/29 genes was increased at 12 h and peaked at 24 h. These results indicated that AtKH9/29 can respond to ABA and SA.

5) In the analysis of seed germination rate, the dose-dependency of ABA and SA and the time-course analysis should be investigated. Without time-course analysis, it is not possible to tell whether seed germination is delayed or does not occur.

Response 5: Thank you so much for your suggestion on our manuscript, the analysis of seed germination rates on the 3 d, 4 d, 6 d, 8 d, 10 d under different concentrations of ABA (0.25 μM, 0.5 μM, 1 μM) and SA (50 μM, 100μM, 200μM) treatment was shown in Figure 8 in the new manuscript. The difference in germination rates between atkh9/29 and wild-type are significant at 6 d then stabilized after 8 d under ABA and SA treatment.

6) Are the genes discussed in Figure 8 genes that are involved in seed germination? Is PR1b a pathogenesis-related gene and has not been reported to be involved in seed germination?

Response 6: Thank you so much for your suggestion on our manuscript, the ABF2, ABF4, P5CS and PR1b all marker genes of the ABA and SA dependent pathway, which aren’t involved in seed germination and mentioned in the paper are used to detect whether kh9/29 responds to ABA and SA stress. So, PR1b isn’t involved in seed germination, which is marker genes of the SA-dependent pathway.

Minor Points

7) In line 19, "inhibit the effect of ABA and SA on seed germination." should be removed.

Response 7: Thank you so much for your suggestion on our manuscript, we have removed these descriptions in the new manuscript.

Reviewer 2 Report

The manuscript ‘Genome-wide characterization and expression analysis of KH family genes response to ABA and SA in Arabidopsis thaliana’ conducted by Zhang et al. identified the KH from published A. thaliana genome. The phylogenetic, synteny analysis with close relatives were included. In addition, the authors applied the deposited RNA-seq data to draw the expression pattern of KH genes under different treatments for seed germination. Finally, two conserved KH genes were included for the functional analysis involved in the seed germination with ABA and SA treatment in A. thaliana mutants. Overall, the component of this work is suitable for publication. However, there are several major and minor concerns can not drive me accept the recent version for IJMS. First, the writing and English is required a substantial revising and editing by proof reading or native English speaker. Second, the arrangement of results should be changed (I suggest it in the following points), and the title of this paper should be changed to more informative and logical. The most serious issue is discussion section. The recent discussion is very weak and most of the content is repeated with ‘results’. I can not see any ‘discussion’ or ‘comparison’ in this section. This section should be rewrote totally, including the KH evolution, and exact function in ABA and SA pathways, cis-elements (if you added), kh9/29 function in seed germination without treatment in previous and present studies., and what we can expect from this study and their potential application?, etc.. Here I list some major and minor issues (just some examples).

  • L14: what’s splicing and transcriptional regulation?
  • L19: sentence wrong, please check: ‘. inhibit the effect of ABA and SA on seed germination.’ As well, the ‘ABA’ and ‘SA’ should be specified in ‘full name’
  • L32 ‘exist’ to ‘exists’
  • L33, similar to point #1
  • Sentence should be revised. For example ‘ interaction regulates xx’, ‘which is the central xxx’.
  • L41 reference is required to support ‘RCF3’ [17] ?
  • L44 ‘is’ to ‘are’
  • L50: response to ABA and SA biosynthesis or regulation, or signal transduction? Should state clearly.
  • Figure 1: Delete ‘Group’ behind ‘Oryza sativa Japonica’. And all the species names should be italic. Under Fig 1, is it possible give the main motif of the identified KH genes.
  • L67: A. thaliana should be italic
  • L68-69: part of the genes are italic, part are not. Please revise.
  • L67-L77 ‘Synteny analysis of KH family’, instead of the synteny pairs analysis between the close relatives, the evolution events probably are more important, for example, gene duplication, gene loss, or others. Here you just mentioned conserved genes. How about others?
  • Figure 2: All the species names should be italic, and give full name of the species
  • L80: at the beginning of sentence, change ‘30’ to ‘Thirty’ or ‘A total of 30’.
  • Section ‘Identification and characterization of KH gene family in Arabidopsis thaliana’ (I revised the subtitle) from L79-97 should move up to the beginning of the ‘Results’. (after L57). This will make the story more logical, from Identification KH from A. thaliana, then phylogenetic analysis with relatives, the synteny analysis.
  • L83: Mw to MW
  • Better to change the decimal of MW in your supplemental table 2, and make consistence with your data interpretation here L83 and L84, ‘KDa’
  • L99-101: should be stated in ‘Materials and Methods’
  • L101 to L104, I will prefer to use ‘expression pattern’ than ‘expression trend’
  • L119-121: should be stated in ‘Materials and Methods’, or you repeated with “Ms&Ms’
  • L126: insert ‘Space’ between (56%) and mutants.
  • L124-L133: A statement about the seed germination in ‘kh9 and kh29’ mutant without any treatment comparing to WT is required. This will give the initial idea about the function of kh9 and kh29. For my observation in your figure 7, it seems that the mutants kh9 and kh29 showed higher seed germination ratio than WT plants (Mock). This may indicate KH9 and KH29 inhibit seed germination in A. thaliana under the non-treatment or application with ABA or SA. Please try to make a clear conclusion of the Kh9/29 function in seed germination.
  • L135-144: Too long statement. Please concise it as the beginning of ‘Discussion’
  • L145-151: This is not informative. And we know that A. thaliana is close to B. rapa. As I suggest above, include the ‘evolution events’ occurred and discuss it may more better.
  • L152-170: most component is repeated with ‘result’. Again, is non informative.
  • L172: change the title
  • L206: N. benthamiana should be expanded with full name
  • L206: manufacture information or confocal microscope. Why not include empty control? Or you have but do not state here?
  • L219: ‘2’ to ‘Two’
  • Can authors conduct the ‘cis-element analysis’ in the promoter region of all the KH genes in A. thaliana.

Author Response

Comments and Suggestions for Authors

The manuscript ‘Genome-wide characterization and expression analysis of KH family genes response to ABA and SA in Arabidopsis thaliana’ conducted by Zhang et al. identified the KH from published A. thaliana genome. The phylogenetic, synteny analysis with close relatives were included. In addition, the authors applied the deposited RNA-seq data to draw the expression pattern of KH genes under different treatments for seed germination. Finally, two conserved KH genes were included for the functional analysis involved in the seed germination with ABA and SA treatment in A. thaliana mutants. Overall, the component of this work is suitable for publication. However, there are several major and minor concerns can not drive me accept the recent version for IJMS. First, the writing and English is required a substantial revising and editing by proof reading or native English speaker. Second, the arrangement of results should be changed (I suggest it in the following points), and the title of this paper should be changed to more informative and logical. The most serious issue is discussion section. The recent discussion is very weak and most of the content is repeated with ‘results’. I can not see any ‘discussion’ or ‘comparison’ in this section. This section should be rewrote totally, including the KH evolution, and exact function in ABA and SA pathways, cis-elements (if you added), kh9/29 function in seed germination without treatment in previous and present studies., and what we can expect from this study and their potential application?, etc.. Here I list some major and minor issues (just some examples).

  1. L14: what’s splicing and transcriptional regulation?

Response 1: Thanks a lot for the reviewers’ suggestion. Splicing and transcriptional regulation are used to describe different biological processes. Splicing (or alternative splicing) is a process in which the exons of the RNA produced by the transcription of the main gene or mRNA precursor are reconnected by RNA splicing in a variety of ways. The resulting different mRNAs may be translated into different protein constructs. Transcription regulation refers to changing the level of gene expression by changing the rate of transcription, which can control when transcription occurs and how much RNA is produced.

  1. L19: sentence wrong, please check: ‘. inhibit the effect of ABA and SA on seed germination.’ As well, the ‘ABA’ and ‘SA’ should be specified in ‘full name’ $

Response 2: Thanks a lot for the reviewers’ suggestion. In line 19, the sentence ‘inhibit the effect of ABA and SA on seed germination.’ were deleted. The ‘ABA’ and ‘SA’ was changed to ‘abscisic acid (ABA)’ and ‘salicylic acid (SA)’.

  1. L32 ‘exist’ to ‘exists’$

Response 3: Thanks a lot for the reviewers’ suggestion. In line 32, the ‘exist’ was changed to ‘exists’

  1. L33, similar to point #1

Response 4: Thanks a lot for the reviewers’ suggestion. In line 33, splicing and transcriptional regulation is used to describe different biological processes. Splicing (or alternative splicing) is a process in which the exons of the RNA produced by the transcription of the main gene or mRNA precursor are reconnected by RNA splicing in a variety of ways. The resulting different mRNAs may be translated into different protein constructs. Transcription regulation refers to changing the level of gene expression by changing the rate of transcription, which can control when transcription occurs and how much RNA is produced.

  1. Sentence should be revised. For example ‘interaction regulates xx’, ‘which is the central xxx’.

Response 5: Thanks a lot for the reviewers’ suggestion. The ‘interact to regulate’ changed to ‘interaction regulates’; the ‘the central’ changed to ‘which is the central’.

  1. L41 reference is required to support ‘RCF3’ [17] ?

Response 6: Thanks a lot for the reviewers’ suggestion. In line 41, reference [17] is added after the ‘RCF3’ .

  1. L44 ‘is’ to ‘are’

Response 7: Thanks a lot for the reviewers’ suggestion. In line 44, the ‘is’ was changed to ‘are’.

  1. L50: response to ABA and SA biosynthesis or regulation, or signal transduction? Should state clearly.

Response 8: Thanks a lot for the reviewers’ suggestion. In line 50, the ‘response to ABA and SA’ was changed to ‘response to ABA and SA stresses’.

  1. Figure 1: Delete ‘Group’ behind ‘Oryza sativa Japonica’. And all the species names should be italic. Under Fig 1, is it possible give the main motif of the identified KH genes.

Response 9: Thanks a lot for the reviewers’ suggestion. ‘Oryza sativa Japonica Group’ was changed to ‘Oryza sativa’ and all species names in manuscript and figures were italic. the main motif of the identified KH genes were shown in Figure 2b in the new manuscript.

  1. L67: thaliana should be italic

Response 10: Thanks a lot for the reviewers’ suggestion. A. thaliana was italic in the new manuscript.

  1. L68-69: part of the genes are italic, part are not. Please revise.

Response 11: Thanks a lot for the reviewers’ suggestion. All genes were italic now in the new manuscript.

  1. L67-L77 ‘Synteny analysis of KH family’, instead of the synteny pairs analysis between the close relatives, the evolution events probably are more important, for example, gene duplication, gene loss, or others. Here you just mentioned conserved genes. How about others?

Response 12: Thanks a lot for the reviewers’ suggestion. We analyzed the synteny within A. thaliana but found there is no tandem duplication and only one segmental duplication in A. thaliana, We had added this description to the discussion in the new manuscript.

  1. Figure 2: All the species names should be italic, and give full name of the species

Response 13: Thanks a lot for the reviewers’ suggestion. All the species names were changed to full name and are italic now, and the Figure 2 was changed to Figure1 in the new manuscript.

  1. L80: at the beginning of sentence, change ‘30’ to ‘Thirty’ or ‘A total of 30’.

Response 14: Thanks a lot for the reviewers’ suggestion.’30 KH family’ was changed to ’ Thirty KH family’.

15.Section ‘Identification and characterization of KH gene family in Arabidopsis thaliana’ (I revised the subtitle) from L79-97 should move up to the beginning of the ‘Results’. (after L57). This will make the story more logical, from Identification KH from A. thaliana, then phylogenetic analysis with relatives, the synteny analysis.

Response 15: Thanks a lot for the reviewers’ suggestion. We had revised the subtitle in the new manuscript.

16.L83: Mw to MW

Response 16: Thanks a lot for the reviewers’ suggestion. The ‘Mw’ was ‘changed to ‘MW’ in the new manuscript.

17.Better to change the decimal of MW in your supplemental table 2, and make consistence with your data interpretation here L83 and L84, ‘KDa’

Response 17: Thanks a lot for the reviewers’ suggestion. We changed the content of the supplemental table 2, and renamed it to Table S1 in the new manuscript.

18.L99-101: should be stated in ‘Materials and Methods’

Response 18: Thanks a lot for the reviewers’ suggestion. We added the description in Materials and Methods in the new manuscript.

19.L101 to L104, I will prefer to use ‘expression pattern’ than ‘expression trend’

Response 19: Thanks a lot for the reviewers’ suggestion.’ had similar trend’ was changed to ‘had similar pattern’ in the new manuscript.

20.L119-121: should be stated in ‘Materials and Methods’, or you repeated with “Ms&Ms’

Response 20: Thanks a lot for the reviewers’ suggestion. We added the description in ‘Materials and Methods’ in the new manuscript.

21.L126: insert ‘Space’ between (56%) and mutants.

Response 21: Thanks a lot for the reviewers’ suggestion. We inserted ‘space' and re-described this part in the new manuscript.

22.L124-L133: A statement about the seed germination in ‘kh9 and kh29’ mutant without any treatment comparing to WT is required. This will give the initial idea about the function of kh9 and kh29. For my observation in your figure 7, it seems that the mutants kh9 and kh29 showed higher seed germination ratio than WT plants (Mock). This may indicate KH9 and KH29 inhibit seed germination in A. thaliana under the non-treatment or application with ABA or SA. Please try to make a clear conclusion of the Kh9/29 function in seed germination.

Response 22: Thanks a lot for the reviewers’ suggestion. The analysis of seed germination rates on the 3 d, 4 d, 6 d, 8 d, 10 d under different concentrations of ABA (0.25 μM, 0.5 μM, 1 μM) and SA (50 μM, 100μM, 200μM) treatment was shown in Figure 8 in the new manuscript. The difference in germination rates between atkh9/29 and wild-type are significant at 6 d then stabilized after 8 d under ABA and SA treatment. the expression of AtKH9/29 genes increased in both treatments, indicating that AtKH9/29 may respond to the regulation of ABA and SA. The marker genes ABF2, ABF4, P5CS, and PR1b of ABA and SA-related were detected, and it was found that the expression level of wild-type was higher than that of atkh9/29 under the same treatment. These results showed that AtKH9 and AtKH29 regulated ABA and SA signaling and promote the effect of ABA and SA on seed germination.

23.L135-144: Too long statement. Please concise it as the beginning of ‘Discussion’

Response 23: Thanks a lot for the reviewers’ suggestion. We had rewritten this part in the new manuscript.

24.L145-151: This is not informative. And we know that A. thaliana is close to B. rapa. As I suggest above, include the ‘evolution events’ occurred and discuss it may more better.

Response 24: Thanks a lot for the reviewers’ suggestion. We had described other evolution events in ‘   Discussion’ in the new manuscript.

25.L152-170: most component is repeated with ‘result’. Again, is non informative.\

Response 25: Thanks a lot for the reviewers’ suggestion. We had rewritten this part in the new manuscript.

26.L172: change the title

Response 26: Thanks a lot for the reviewers’ suggestion. We changed the title ’Identification and characteristics of KH family genes’ to ‘Identification and characterization of KH gene family in Arabidopsis thaliana’.

27.L206: N. benthamiana should be expanded with full name

Response 27: Thanks a lot for the reviewers’ suggestion. We changed the ‘N. benthamiana’ to ‘Nicotiana benthamiana’.

28.L206: manufacture information or confocal microscope. Why not include empty control? Or you have but do not state here?

Response 28: Thanks a lot for the reviewers’ suggestion. In Figure 6 at the first row, empty control was described as ‘35:GFP’.

29.L219: ‘2’ to ‘Two’

Response 29: Thanks a lot for the reviewers’ suggestion. We changed the ’Two days later’ to ’Two days later’.

30.Can authors conduct the ‘cis-element analysis’ in the promoter region of all the KH genes in A. thaliana.

Response 30: Thanks a lot for the reviewers’ suggestion. We analyzed the cis-element in the promoter region of KH family in A. thaliana and described it in ‘Result’ and ‘Discussion’.

Round 2

Reviewer 1 Report

The revised manuscript has improved with the additional experiments suggested in the previous review. The reviewer suggests a few minor issues that should be considered before accepting this manuscript.

1) Figure S3 regarding synteny blocks needs more explanation in the manuscript. The figure legend needs to include a description of the line colors, and that blue line is the KH gene.

2) The germination test has improved with the analysis of dose-dependency and time-course. In Fig. 8b and c, the maximum vertical axis (%)value is 1.0, but should it be 100? In Fig.8a, it looks like most of the seeds have germinated. Also, the image in fig. 8A is complicated, so consider moving it to the Supplemental Figure.

Author Response

Comments and Suggestions for Authors

The revised manuscript has improved with the additional experiments suggested in the previous review. The reviewer suggests a few minor issues that should be considered before accepting this manuscript.

1) Figure S3 regarding synteny blocks needs more explanation in the manuscript. The figure legend needs to include a description of the line colors, and that blue line is the KH gene.

Response 1: Thank you so much for your suggestion on our manuscript, we described the Figure S3 in title of Figure S3: ‘Figure S3: An example of AtKH9-BrKH1 to display synteny blocks, the blue line refers to AtKH9-BrKH1, and the other colored lines refer to the surrounding genes of AtKH9 and BrKH1. ’ in our new manuscript. Also, we added a figure legend to Figure S3.

  • The germination test has improved with the analysis of dose-dependency and time-course. In Fig. 8b and c, the maximum vertical axis (%)value is 1.0, but should it be 100? In Fig.8a, it looks like most of the seeds have germinated. Also, the image in fig. 8A is complicated, so consider moving it to the Supplemental Figure.

Response 2: Thank you so much for your suggestion on our manuscript, we changed the maximum vertical axis (%) value of Figure 8b to 100 in our new manuscript. We moved the Figure 8 to Supplemental Figures as Figure S4, then add the plot of seed germination rates of atkh9 and atkh29 under ABA and SA treatments as the new Figure 8 in our new manuscript.

Reviewer 2 Report

First, I appreciate of the author did an extensive revising and editing on the first version of this manuscript. Now the manuscript was improved. However, I do still feel that the author should take serious about the manuscript preparation. Even most of these are minor editing or writing issue, however, as a scientific writing we should act professional in publishing certain work. This let me hard to accept the manuscript for publication in current form again. Here I just list you some examples (REMEMBER: EXAMPLES) and you have to go through it carefully word by word. 

EXAMPLES:
1)L54 & L57: the names of ABA and SA should be expanded earlier.  You state abbreviation ABA and SA first and then expand them later. 

2) L55: 'ABF' and 'PSC5' stand for:

3) Insert space between 'number' and ' unit' for example L81, others (please go through the manuscript carefully)

4) L102 'response to ABA and SA stresses': usually ABA and SA are not stress. They are either treatment or function in signal transduction. 

6) L106: if you indiate AtKH1 to AtKH30 as gene, you have to italise them

7) L112-113 repeated with the 'Materials and Methods' and please rephrase it

8) L116 subgroups or subgroup?

9) L118-119: 'amount' or 'number'?

10) L122: The writing of  Roma 'II' is not consistence in the manuscript.

11) L123, L125, : 'cis' should be italic

12)  L135: Change '6' to 'six' (usually in small number, better to spell it out)

13) L134-136: this is 'materials and methods'

14) L138-139: please make this sentence informative. You can not always request readers to read by themselves. You have to highlight main finding in this figure.

15) AtKH9 and AtKH29 were supposed to be more conserved because of their presence in more of investigated plant species. You may want to give a definition or citation when you describe 'Materials and Methods' about this point.

16) L189" 'flowering regulation'? can you specify it to 'flowering time' or 'flower development' or 'others'

17) L190-193: Please describe what's exactly 'trend'? increase, decrease, or increase first and then decrease.  Do not always use the 'similar trend'. This is non informative.

18) L196-197: repeat with 'Materials and Methods'

19) L202: 'Changed' ?? What's kind of change?

20) L211-214 : repeat with 'Materials and Methods'. Please rephrase this paragraph.

21) L222: please check the 'space' between 'word' and 'period'

22) L224-225: repeat with 'Materials and Methods'

23) L227: 'stresses' or 'treatments'

24) L227-228: what are the 'differences'

25) L232: 'P5CS'

26) L231-233: please carefully check, there references were mentioned in 'Materials and Methods'. Not only 'cite' here in result section.

27) "accidentally'?? why 'accidentally'

28) L243 'cis' should be in italic

29) L245: 30 KH family genes or 30 KH family (families)? ; change 'including' to 'includes'

30) L246: 'cis' 'A. thaliana' should be italic.

31) L249 &250:  'cis'

32) L307" 'AtKH9 and AtKH29' 

33) L310-313: the reason 'due to different conditions' should be detailed. 

34) Overall, the 'discussion' section from L237-322 should be written in depth. Current, the discussion is not sufficient.

35) L339:  'Mw'

36) L343 'cis'

37) L356 '5' to 'five'

38) L372: 'MgCl2'

39) L377: subtitle should change. or merge with the 'Seed Germination Rate?'

40) L384: ' Seed germination rate'? how about 'Seed germination' 

41) L390: 'grains' or 'seeds'

42) L393: ABA and SA-related Gene Expression Analysis

43) L 394-404: Please indicate the information of investigated 'ABA/SA' genes in this section' NOT ONLY JUST CITE IN 'RESULT' SECTION. L404, you have to mention that the 'ABA/SA' related gene and the rational use of them.

AGAIN, above are some EXAMPLES ONLY.

Other issues:

The manuscript prepared is not followed the Journal standard.

The format of references are NOT meet the journal requirement and NOT consistence. 

Author Response

Comments and Suggestions for Authors

First, I appreciate of the author did an extensive revising and editing on the first version of this manuscript. Now the manuscript was improved. However, I do still feel that the author should take serious about the manuscript preparation. Even most of these are minor editing or writing issue, however, as a scientific writing we should act professional in publishing certain work. This let me hard to accept the manuscript for publication in current form again. Here I just list you some examples (REMEMBER: EXAMPLES) and you have to go through it carefully word by word.

EXAMPLES:

1)L54 & L57: the names of ABA and SA should be expanded earlier. You state abbreviation ABA and SA first and then expand them later.

Response 1: Thanks a lot for the reviewers’ suggestion. We changed 'ABA' and 'SA' to 'abscisic acid (ABA)' and 'salicylic acid (SA)' respectively.

2) L55: 'ABF' and 'PSC5' stand for:

Response 2: Thanks a lot for the reviewers’ suggestion. We changed 'ABF2' and 'PSC5' to 'ABRE-binding factor 2 (ABF2)' and 'delta 1-pyrroline-5-carboxylate synthase (P5CS)' respectively.

3) Insert space between 'number' and ' unit' for example L81, others (please go through the manuscript carefully)

Response 3: Thanks a lot for the reviewers’ suggestion. We changed '16℃' to '16 ℃', and inserted space between 'number' and ' unit' in full text of manuscript.

4) L102 'response to ABA and SA stresses': usually ABA and SA are not stress. They are either treatment or function in signal transduction.

Response 5: Thanks a lot for the reviewers’ suggestion. We changed ‘response to ABA and SA stresses' to 'response to ABA and SA treatment'.

6) L106: if you indiate AtKH1 to AtKH30 as gene, you have to italise them

Response 6: Thanks a lot for the reviewers’ suggestion. We transformed all genes into italics.

7) L112-113 repeated with the 'Materials and Methods' and please rephrase it

Response 7: Thanks a lot for the reviewers’ suggestion. We changed this part to ‘A phylogenetic tree was constructed to analysis the possible evolutionary relationship of KH family in A. thaliana.’

8) L116 subgroups or subgroup?

Response 8: Thanks a lot for the reviewers’ suggestion. We changed ‘subgroups’ to 'subgroup'.

9) L118-119: 'amount' or 'number'?

Response 9: Thanks a lot for the reviewers’ suggestion. We changed ‘amount’ to 'number'.

10) L122: The writing of  Roma 'II' is not consistence in the manuscript.

Response 10: Thanks a lot for the reviewers’ suggestion.We adjusted the font of all Roman letters in the manuscript

11) L123, L125, : 'cis' should be italic

Response 11: Thanks a lot for the reviewers’ suggestion. We changed ‘cis-element’ to ‘cis-element’.

12)  L135: Change '6' to 'six' (usually in small number, better to spell it out)

Response 12: Thanks a lot for the reviewers’ suggestion. We changed ‘6 species’ to ‘six species’.

13) L134-136: this is 'materials and methods'

Response 13: Thanks a lot for the reviewers’ suggestion. We changed this part to ‘All of these 241 genes were used to construct the phylogenetic tree of the KH family. ’

14) L138-139: please make this sentence informative. You can not always request readers to read by themselves. You have to highlight main finding in this figure.

Response 14: Thanks a lot for the reviewers’ suggestion. We changed this part to ‘ A total of 233 of these 241 genes were found to contain similar motif sequences with a high proportion of G in 19th and 22nd positions in the KH motif sequence (GEV-TVRJLVPSSKVGSIIGKGGSTIKRJREETGARIRI), which may be the conserved se-quence motif in KH domain [27] (Figure 2b).’

15) AtKH9 and AtKH29 were supposed to be more conserved because of their presence in more of investigated plant species. You may want to give a definition or citation when you describe 'Materials and Methods' about this point.

Response 15: Thanks a lot for the reviewers’ suggestion. We described this part in ‘Materials and Methods’ as ‘Our bioinformatic analysis shows that Atkh9 and Atkh29 are more conservative compared to other KH family genes’

16) L189" 'flowering regulation'? can you specify it to 'flowering time' or 'flower development' or 'others'

Response 16: Thanks a lot for the reviewers’ suggestion. We changed ‘flowering regulation’ to ‘flower development regulation’.

17) L190-193: Please describe what's exactly 'trend'? increase, decrease, or increase first and then decrease.  Do not always use the 'similar trend'. This is non informative.

Response 17: Thanks a lot for the reviewers’ suggestion. We changed this part to ‘Interestingly, we found that with the growth and development of plants, the expression level of AtKH12 will continue to increase and AtKH9 will continue to decrease though both of them were included in the same subgroup. In contrast, AtKH3 and AtKH9 showed the highest expression levels in the seed stage, decreased expression levels in the leaf stages (relatively consistent expression levels in the leaf stage 1 and leaf stage 2), and further decreased in the flower stage and senescence stage though they belonged to two different subgroups. In addition, the other 15 genes had the highest expression in the seed stage.’

18) L196-197: repeat with 'Materials and Methods'

Response 18: Thanks a lot for the reviewers’ suggestion. We changed this part to ‘Expression data of A. thaliana was downloaded to analyze changes of expression under ABA and SA treatment.’

19) L202: 'Changed' ?? What's kind of change?

Response 19: Thanks a lot for the reviewers’ suggestion. We changed ‘AtKH8/14/27 was significantly changed’ to ‘AtKH8/14/27 was significantly changed’.

20) L211-214 : repeat with 'Materials and Methods'. Please rephrase this paragraph.

Response 20: Thanks a lot for the reviewers’ suggestion. We changed this part to ‘To further verify the function the of KH family, we selected the most conserved two genes in subgroup â… , AtKH9/29 for the next analysis. ’

21) L222: please check the 'space' between 'word' and 'period'

Response 21: Thanks a lot for the reviewers’ suggestion. We deleted the space between ‘Figure 7’ and ‘.’.

22) L224-225: repeat with 'Materials and Methods'

Response 22: Thanks a lot for the reviewers’ suggestion. We changed this part to ‘In order to further determine the functions of AtKH9 and AtKH29, atkh9 and atkh29 mutants were sown containing ABA and SA with different concentrations.’

23) L227: 'stresses' or 'treatments'

Response 23: Thanks a lot for the reviewers’ suggestion. We changed ‘ABA and SA stresses’ to ‘ABA and SA treatments’.

24) L227-228: what are the 'differences'

Response 24: Thanks a lot for the reviewers’ suggestion. We changed ‘ The difference in germination rates between atkh9/29 and wild-type were significant at 6 d then stabilized after 8 d under ABA and SA treatment’ to ‘The germination rates of atkh9/29 and wide-type under ABA and SA treatments changed most obviously on the 6 d, and they remained stable on the 8 d (Figure S4)’. The ‘difference’ is the difference in the germination rate of all samples on the 6 d and the 8 d.

25) L232: 'P5CS'

Response 25: Thanks a lot for the reviewers’ suggestion. We changed ‘P5CS’ to ‘delta 1-pyrroline-5-carboxylate synthase (P5CS)’

26) L231-233: please carefully check, there references were mentioned in 'Materials and Methods'. Not only 'cite' here in result section.

Response 26: Thanks a lot for the reviewers’ suggestion. We described this part in ‘Materials and Methods’ as ‘The cDNA was used to determine the expression of ABA and SA-related genes ABF2, ABF4, P5CS, PR1b [28-31].’

27) "accidentally'?? why 'accidentally'

Response 28: Thanks a lot for the reviewers’ suggestion. We deleted this word.

28) L243 'cis' should be in italic

Response 28: Thanks a lot for the reviewers’ suggestion. We changed ‘cis-element’ to ‘cis-element’.

29) L245: 30 KH family genes or 30 KH family (families)? ; change 'including' to 'includes'

Response 29: Thanks a lot for the reviewers’ suggestion. We changed ‘30 KH family ’ to ‘30 KH family genes’ and rewritten the ‘Discussion’.

30) L246: 'cis' 'A. thaliana' should be italic.

Response 30: Thanks a lot for the reviewers’ suggestion. We changed ‘cis-element’ and ‘A. thaliana’ to ‘cis-element’ and ‘A. thaliana’.

31) L249 &250:  'cis'

Response 31: Thanks a lot for the reviewers’ suggestion. We changed ‘cis-element’ to ‘cis-element’.

32) L307" 'AtKH9 and AtKH29'

Response 32: Thanks a lot for the reviewers’ suggestion. We changed ‘AtKH9 and AtKH29’ to ‘AtKH9 and AtKH29’ .

33) L310-313: the reason 'due to different conditions' should be detailed.

Response 33: Thanks a lot for the reviewers’ suggestion. We deleted the discussion about this part.

34) Overall, the 'discussion' section from L237-322 should be written in depth. Current, the discussion is not sufficient.

Response 34: Thanks a lot for the reviewers’ suggestion. We rewritten the ‘Discussion’

35) L339:  'Mw'

Response 35: Thanks a lot for the reviewers’ suggestion. We changed ‘Mw’ to ‘MW’ .

36) L343 'cis'

Response 36: Thanks a lot for the reviewers’ suggestion. We changed ‘cis-element’ to ‘cis-element’.

37) L356 '5' to 'five'

Response 37: Thanks a lot for the reviewers’ suggestion. We changed ‘divided into 5 stages’ to ‘divided into five stages’.

38) L372: 'MgCl2'

Response 38: Thanks a lot for the reviewers’ suggestion. We changed ‘MgCl2’ to ‘MgCl2’.

39) L377: subtitle should change. or merge with the 'Seed Germination Rate?'

Response 39: Thanks a lot for the reviewers’ suggestion. We changed the subtitle to ‘Materials of AtKH9/29 genes expression analysis and seed germination’.

40) L384: ' Seed germination rate'? how about 'Seed germination'

Response 40: Thanks a lot for the reviewers’ suggestion. We merged ‘Materials of AtKH9/29 genes expression analysis’ and ‘Seed germination rate’ to ‘Materials of AtKH9/29 genes expression analysis and seed germination’.

41) L390: 'grains' or 'seeds'

Response 41: Thanks a lot for the reviewers’ suggestion. We changed ‘54 grains’ to  ‘54 seeds’.

42) L393: ABA and SA-related Gene Expression Analysis

Response 42: Thanks a lot for the reviewers’ suggestion. We changed ‘Genes expression analysis of ABA and SA-related’ to ‘ABA and SA-related Gene Expression Analysis’.

43) L 394-404: Please indicate the information of investigated 'ABA/SA' genes in this section' NOT ONLY JUST CITE IN 'RESULT' SECTION. L404, you have to mention that the 'ABA/SA' related gene and the rational use of them.

Response 43: Thanks a lot for the reviewers’ suggestion. We described this part in ‘Materials and Methods’ as ‘The cDNA was used to determine the expression of ABA and SA-related genes ABF2, ABF4, P5CS, PR1b [28-31].’

AGAIN, above are some EXAMPLES ONLY.

Other issues:

The manuscript prepared is not followed the Journal standard.

Response 44: Thanks a lot for the reviewers’ suggestion. We adjusted our manuscript to make it followed the Journal standard.

The format of references are NOT meet the journal requirement and NOT consistence.

Response 45: Thanks a lot for the reviewers’ suggestion. We adjusted our references to make it followed the Journal requirement.

Round 3

Reviewer 2 Report

I hope you did a carefully reading throughout the manuscript not only limit in the examples I given. 

Thanks for your effort to revise the manuscript.